# Study environment and the incidence of mental health problems and activity-limiting musculoskeletal problems among university students: the SUN cohort study

Fred Johansson ®,[1] Jessica Billquist,[2] Hanna Andreasson,[2] Irene Jensen,[3] Clara Onell,[1] Anne H Berman ®,[2,4] Eva Skillgate[1,3,5]

For numbered affiliations see end of article.

**Correspondence to**
Fred Johansson;
fred.johansson@shh.se

## ABSTRACT

**Objective** To determine the association between different aspects of study environment and the incidence of mental health problems and activity-limiting musculoskeletal problems.

**Design, setting and participants** We recruited a cohort of 4262 Swedish university students of whom 2503 (59%) were without moderate or worse mental health problems and 2871 (67%) without activity-limiting musculoskeletal problems at baseline. The participants were followed at five time points over 1 year using web surveys.

**Exposures** Self-rated discrimination, high study pace, low social cohesion and poor physical environment measured at baseline.

**Outcomes** Self-rated mental health problems defined as scoring above cut-off on any of the subscales of the Depression, Anxiety and Stress Scale. Self-rated activity-limiting musculoskeletal problems in any body location assessed by the Nordic Musculoskeletal Questionnaire.

**Statistical analysis** Discrete survival-time analysis was used to estimate the hazard rate ratio (HR) of each exposure–outcome combination while adjusting for gender, age, living situation, education type, year of studies, place of birth and parental education as potential confounders.

**Results** For discrimination, adjusted HRs were 1.75 (95% CI 1.40 to 2.19) for mental health problems and 1.39 (95% CI 1.12 to 1.72) for activity-limiting musculoskeletal problems. For high study pace, adjusted HRs were 1.70 (95% CI 1.48 to 1.94) for mental health problems and 1.25 (95% CI 1.09 to 1.43) for activity-limiting musculoskeletal problems. For low social cohesion, adjusted HRs were 1.51 (95% CI 1.29 to 1.77) for mental health problems and 1.08 (95% CI 0.93 to 1.25) for activity-limiting musculoskeletal problems. For perceived poor physical study environment, adjusted HRs were 1.20 (95% CI 0.99 to 1.45) for mental health problems and 1.20 (95% CI 1.01 to 1.43) for activity-limiting musculoskeletal problems.

**Conclusions** Several aspects of the study environment were associated with the incidence of mental health

---

## STRENGTHS AND LIMITATIONS OF THIS STUDY

⇒ We recruited a large sample of students, which enabled us to estimate associations between the exposures and the outcomes with good precision.
⇒ We followed the participants over time and were able to adjust for multiple potential confounders, which strengthens the evidence level for causal associations.
⇒ There may still be residual and unmeasured confounding and selection bias that could affect the associations.
⇒ Exposures were measured with single items questions, which could increase the risk of misclassification.
⇒ The sample was measured mostly during the COVID-19 pandemic and is not representative of Swedish university students overall, which may affect generalisability.

---

problems and activity-limiting musculoskeletal problems in this sample of Swedish university students.

## INTRODUCTION

Mental health problems and musculoskeletal problems, such as neck and back pain, are leading causes of years lost due to disability in young people globally,[1] and common among university students.[2–5] Previous studies show that 35.7% of students from high-income countries meet criteria for at least one common mental disorder.[5] Mental health problems are associated with increased risk of student attrition,[5] academic impairment,[6] reduced occupational preparedness and future work performance.[7] Musculoskeletal pain is also common among university students[8–10] and associated with poorer academic functioning and lower quality of life.[11]

Mental health problems and pain commonly co-occur.[12] This comorbidity has been suggested to be characterised by bidirectional causation, with pain influencing mental health and vice versa.[13 14] Given the high comorbidity and bidirectional influences, it is likely that these conditions share many risk factors. Some risk factors are likely to affect students and non-students alike. Others, such as the exposure to university study environments, are specific to students and pose an important research target for understanding and preventing student health problems. Study environment is a multifaceted construct with several physical and psychosocial aspects. This study focuses on four specific aspects: discrimination, study pace, social cohesion and physical study environment.

Seven per cent of Swedish students report experiencing discrimination during their studies.[15] Perceived discrimination is associated with mental and physical ill health[16] and is negatively associated with subsequent psychological well-being even after controlling for baseline levels of psychological well-being.[17] Brown et al[18] showed that the association between discrimination and chronic pain was mediated through psychological distress in a general population sample. However, little is known as to how discrimination occurring specifically at the university affects students' health.

Perceived stress and academic demands have been associated with higher levels of depressive symptoms and psychological distress among students in longitudinal studies,[19 20] and perceived work/study demands with neck and upper back pain among young adults.[21] However, few studies have focused specifically on high study pace and its associations to mental health and activity-limiting musculoskeletal problems.

Social isolation and loneliness are well-established risk factors for mental health problems in general,[22] but less is known about the impact of low social cohesion within the study environment. For adolescents, a lack of social cohesion at school and lack of classmate support are cross-sectionally associated with lower psychological well-being and higher symptom levels of depression and anxiety.[23] Low workplace social support has also been identified as a risk factor for back pain among working adults,[24] but to our knowledge, there are no studies on the association between social cohesion and activity-limiting musculoskeletal problems among university students.

Physical aspects of the study environment such as repetitive movement, prolonged sitting and sitting position are associated with musculoskeletal pain among university students[25] and school children.[26] However, there is a lack of research into how students' perceptions of the physical study environment is associated with subsequent mental health problems and activity-limiting musculoskeletal problems.

In this study, we aimed to determine the associations between four aspects of study environment: discrimination, high study pace, low social cohesion and perceived poor physical environment, and the incidence of mental health problems and activity-limiting musculoskeletal problems among Swedish university students.

## METHOD

### Design and participants

This study is based on the Sustainable UNiversity life (SUN) cohort,[27] a longitudinal study aiming to identify risk and prognostic factors for mental health problems and musculoskeletal pain and problems in university students. Participants were recruited between August 2019 and December 2020, from eight universities in the greater Stockholm area and Örebro. The targeted universities constitute a convenience sample, selected to cover a variety of educational programmes mainly within medicine, business, technology and social sciences. Full-time undergraduate or graduate students (up to and including masters' level) at selected educational programmes with at least one full year of academic studies left before graduation were eligible for inclusion.

Eligible students were informed about the study during an in-class presentation and/or by email with a link to the baseline survey. Students agreeing to participate were followed every 3 months for 1 year using web surveys. For the aim of studying incidence of mental health problems and activity-limiting musculoskeletal problems in this study, two cohorts were established: (1) the mental health cohort (MHC) including participants without moderate or more severe mental health problems at baseline and (2) the musculoskeletal health cohort (MSKHC) including participants without activity-limiting musculoskeletal problems at any body location at baseline (figure 1).

More information about the study methodology and data collection is available in the study protocol.[27]

### Patient and public involvement

We involved students by conducting focus group interviews on health challenges of university students during the planning phase of the study. We have also had an ongoing dialogue with student health services, educational offices and student unions during the planning and conduct of the study, as well as for dissemination of the results. Students and university officials have been involved in the selection of exposures and outcomes, in recruitment and data collection phase, and some have participated in seminars on the results, organised by the research group.

### Measures

#### Outcomes

Mental health problems were measured with the short-form Depression, Anxiety, and Stress Scale (DASS-21),[28] consisting of 21 items including subscales on depression, anxiety and stress. Each subscale comprises seven items rated on a scale from 0 (did not apply to me at all) to 3 (applied to me very much, or most of the time), with a recall time of 1 week. The items were summed to give subscale scores ranging from 0 to 21. Participants scoring

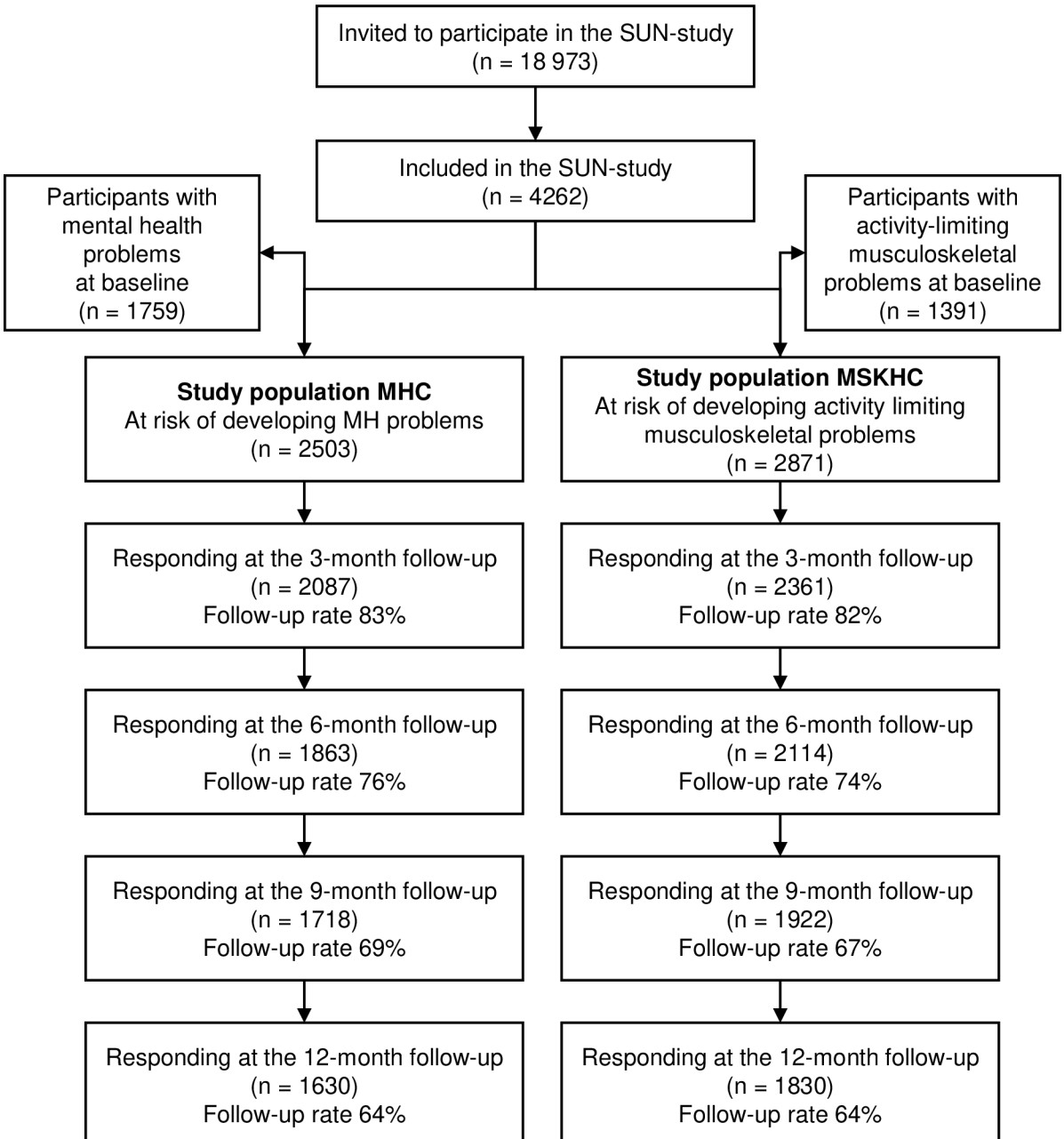

**Figure 1** Flow chart of the inclusion of participants. MHC, mental health cohort; MSKHC, musculoskeletal health cohort; SUN, Sustainable UNiversity Life.

above the cut-off for moderate symptoms levels on any of the scales (≥7 on the depression subscale, ≥6 on the anxiety subscale, ≥10 on the stress subscale) were considered to have mental health problems. DASS-21 has shown good psychometric properties among Swedish university students.[29]

Activity-limiting musculoskeletal problems were measured with the Nordic Musculoskeletal Questionnaire.[30] Participants were asked if they 'at any time during the last 3 months have been prevented from doing your normal activities (eg, work, studies, household activities, hobbies) due to trouble with:' ankles/feet, knees, hips, lower back, upper back, shoulders, elbows, hands and/ or neck. The question was responded to dichotomously

(Yes/No) for each body location. Participants responding 'Yes' for any of the body locations were considered to have activity-limiting musculoskeletal problems.

### Exposures

All exposures were measured at baseline.

Discrimination was measured by the question: 'Have you ever during the past year felt discriminated by other students, teachers or administrative personnel due to:' over nine categories: gender, gender identity, gender expression, ethnicity, religion or beliefs, sexual orientation, disabilities, age, other.[15] Participants answering 'Yes' on at least one of the categories were asked to rate how severely the discrimination had affected them on a

scale from 0 (not at all) to 10 (worst imaginable way). Participants with severity ratings from the 50th percentile (>3) were considered exposed. This cut-off was decided a priori to identify more severe forms of discrimination.

Study pace was measured by the questions: 'Do you feel that you are coping with the study pace?' with three response categories: (1) 'I feel that I have energy and time to cope with the study pace', (2) 'The study pace is high. I do my best, but I am afraid that it is not enough' and (3) 'The study pace breaks me. We get more assignments than I have time for, and I feel that I cannot make it'. Participants choosing categories 2 or 3 were considered exposed to high study pace.

Social cohesion was measured by the question: 'How good is the cohesion with your course mates?', with four response alternatives: (1) 'The cohesion is pretty good. We help each other with the studies and there is always someone in the class to talk to', (2) 'I do not think that our cohesion is good. We work independently, and it feels like it is not worth the inconvenience to try to change it', (3) 'We do not have any cohesion, and no one takes any responsibility over our groupwork or the fellowship' and (4) 'My education is not organised around study groups or classes'. Participants choosing categories 2 or 3 were considered exposed to low social cohesion. Participants choosing the fourth category were excluded from the risk analyses.

Perceived physical environment was measured by asking: 'How do you in general experience the physical study environment at your university? (eg, public areas, lunchroom, lecture halls, toilets, outside environment around campus.)' and rated on a scale from 0 (very bad) to 10 (very good). Participants with ratings of 0–5 were considered exposed to a poor physical study environment.

### Covariates

Potential confounders were selected based on previous research[4 31–33] and directed acyclic graphs.[34] We included the covariates gender (female, male, other), age (continuous), living situation (alone, with others), education type (medical/health, technical, social sciences/humanities, economic, other), year of studies (first, second, above), place of birth (Sweden, Scandinavia, Europe, outside Europe) and highest level of parental education (university, below university) as a proxy for socioeconomic status.

### Statistical analysis

We used discrete-time survival analysis to estimate associations between baseline exposures and the hazard of each of the outcomes. A participant was considered at risk for the outcome until the outcome was first registered or until censoring due to lost to follow-up or end of follow-up, whichever came first.

We used generalised linear models with binomial error distributions and complementary log-log links to estimate the hazard rate ratio (HR) with 95% confidence intervals (CI) of the outcomes comparing the exposed to the unexposed. Separate models were built for each exposure–outcome combination. This modelling approach is equivalent to Cox proportional hazard models when assuming that the underlying time variable is continuous.[35] Time was measured as four time periods (0–3 months, 3–6 months, 6–9 months and 9–12 months). Plotting the log hazard over the four time periods showed a fairly linear relationship for both outcomes, and therefore, the time period variable was treated as continuous. The proportional hazard assumption was evaluated by including interaction terms between time period and the exposures and performing likelihood ratio tests comparing models with and without interaction terms. These indicated better fits for models assuming proportional hazards for all exposure–outcome combinations, except for the model estimating the association between discrimination and mental health problems.

### Sensitivity analyses

Several additional sensitivity analyses were run as detailed below. First, subclinical symptom levels of mental health problems at baseline were assumed to be on the causal pathway and thus not included as covariates in the main analyses. To assess the sensitivity of our analyses to this assumption, we performed sensitivity analyses where the continuous scores of the three DASS-21 subscales, measured at baseline, were included as covariates. Second, data collection took place before and during the COVID-19 pandemic, which may have affected both our exposures and outcomes thus potentially leading to confounding. About half of the participants entered the SUN-study before the COVID-19 pandemic started in Sweden on 13 March 2020 (45% in MHC and 41% in MSKHC), and almost all outcome data were collected after the onset of the pandemic (92% in the MHC and 83% in the MSKHC). To assess the robustness of our results to potential confounding by the COVID-19 pandemic we, reran the main analyses while including a covariate indicating if the baseline survey, where exposures were measured, was collected before or after 13 March 2020. Third, to assess potential effect modification by gender, gender-stratified risk analyses were performed. Fourth, to assess potential selection bias we computed risk ratios (RR) of lost to follow-up at the 12-month follow-up between exposed and unexposed,[36] and computed E-values defining the minimum association between missingness and the outcome, on the RR scale, needed to shift the point estimates of the main analysis to the null.[37] Fifth, to investigate potential dose–response relationships for the exposure's high study pace and low social cohesion we performed sensitivity analyses treating each response category as a separate exposure level.

All analyses were performed using R V.4.1.2.

## RESULTS

We included 4262 participants, of whom 2503 (59%) were without moderate or worse mental health problems at baseline and were included in the MHC and 2871

(67%) were without activity-limiting musculoskeletal problems in any body location at baseline and included in the MSKHC. A flow chart of inclusion of study participants and follow-up rates is presented in figure 1. Only participants responding to at least one follow-up could be included in the analytic sample for the main analysis, giving a total of 2200 participants in the MHC, and 2513 in the MSKHC.

In the MHC, the mean age was 25 years and 57% were females. Seven per cent had been exposed to discrimination, 47% experienced a high study pace, 19% experienced low social cohesion and 13% perceived the physical study environment as poor (table 1). In the MSKHC, the mean age was 24.5 years and 59% were female. Eight per cent had been exposed to discrimination, 55% experienced a high study pace, 24% experienced low social cohesion and 15% perceived the physical study environment as poor (table 1). Characteristics of all study participants stratified by exposure status is presented in tables 2 and 3.

### Risk analysis
The number of outcome events at each time period is presented in online supplemental eTable 1. When comparing the hazard of the outcomes between exposed and unexposed participants, discrimination gave adjusted HRs of 1.75 (95% CI 1.40 to 2.19) for moderate or worse mental health problems and 1.39 (95% CI 1.12 to 1.72) for activity-limiting musculoskeletal problems in any body location. High study pace gave adjusted HRs of 1.70 (95% CI 1.48 to 1.94) for mental health problems and 1.25 (95% CI 1.09 to 1.43) for activity-limiting musculoskeletal problems. Low social cohesion gave adjusted HRs of 1.51 (95% CI 1.29 to 1.77) for mental health problems and 1.08 (95% CI 0.93 to 1.25) for activity-limiting musculoskeletal problems. Poor physical environment gave adjusted HRs of 1.20 (95% CI 0.99 to 1.45) for mental health problems, and 1.20 (95% CI 1.01 to 1.43) for activity-limiting musculoskeletal problems (table 4).

### Sensitivity analyses
Additionally adjusting for baseline levels of depression, anxiety and stress symptoms gave the following HRs for mental health problems: discrimination (HR 1.34 (95% CI 1.07 to 1.69)), high study pace (HR 1.18 (95% CI 1.02 to 1.36)), low social cohesion (HR 1.28 (95% CI 1.09 to 1.50)), poor physical environment (HR 1.02 (95% CI 0.84 to 1.24)). For the outcome activity-limiting musculoskeletal problems, the HRs adjusted for baseline levels of depression, anxiety and stress symptoms were: discrimination (HR 1.28 (95% CI 1.03 to 1.60)), high study pace (HR 1.13 (95% CI 0.98 to 1.30)), low social cohesion (HR 1.01 (95% CI 0.87 to 1.18)), poor physical environment (HR 1.15 (95% CI 0.97 to 1.37)). Adding a variable indicating if exposures were measured before or during the COVID-19 pandemic did not affect the main results (online supplemental eTable 2, table 4). Gender-stratified analyses showed some potential differences in

the strength of the associations between men and women for mental health, although differences are uncertain due to overlapping CIs. The point estimates for low social cohesion and high study pace were higher among men (online supplemental eTable 3). Sensitivity analysis for selection bias showed that all exposures were weakly related to lost to follow-up (online supplemental eTable 4). The E-value analysis indicated that for some exposures, relatively weak associations between the outcome and missingness could affect the external validity of the

**Table 1** Characteristics of the participants in the two cohorts

| | Mental health cohort (n=2503) | Musculoskeletal health cohort (n=2871) |
|---|---|---|
| Age, mean (SD) | 25.0 (6.4) | 24.5 (6.1) |
| Gender, n (%) | | |
| Female | 1427 (57) | 1698 (59) |
| Male | 1065 (43) | 1157 (40) |
| Other | 11 (0) | 16 (1) |
| Living status, n (%) | | |
| Alone | 695 (28) | 806 (28) |
| With others | 1808 (72) | 2065 (72) |
| Education type, n (%) | | |
| Medical/health | 1184 (47) | 1271 (44) |
| Technical | 989 (40) | 1231 (43) |
| Social sciences/humanities | 228 (9) | 243 (9) |
| Economic | 59 (2) | 86 (3) |
| Other | 43 (2) | 40 (1) |
| Highest parental education, n (%) | | |
| University | 1826 (73) | 2097 (73) |
| Below university | 677 (27) | 774 (27) |
| Study year, n (%) | | |
| First year | 1032 (41) | 1207 (42) |
| Above first year | 1471 (59) | 1664 (58) |
| Place of birth, n (%) | | |
| Sweden | 1995 (80) | 2247 (78) |
| Scandinavia | 90 (4) | 95 (3) |
| Europe | 151 (6) | 172 (6) |
| Outside Europe | 267 (11) | 357 (12) |
| Discrimination, n (%) | 174 (7) | 239 (8) |
| High study pace, n (%) | 1180 (47) | 1587 (55) |
| Low social cohesion, n (%) | 479 (19) | 690 (24) |
| Poor physical study environment, n (%) | 327 (13) | 437 (15) |

**Table 2** Characteristics of the participants in the mental health cohort by exposure status

| | Discrimination | | High study pace | | Low social cohesion* | | Poor physical environment | |
|---|---|---|---|---|---|---|---|---|
| | Unexposed (n=2329) | Exposed (n=174) | Unexposed (n=1323) | Exposed (n=1180) | Unexposed (n=1986) | Exposed (n=479) | Unexposed (n=2176) | Exposed (n=327) |
| Age, mean (SD) | 24.9 (6.3) | 25.9 (7.1) | 25.2 (6.5) | 24.8 (6.4) | 24.7 (6.2) | 25.4 (6.6) | 24.8 (6.3) | 26.5 (7.1) |
| Gender, n (%) | | | | | | | | |
| Female | 1301 (56) | 126 (72) | 680 (51) | 747 (63) | 1145 (58) | 254 (53) | 1231 (57) | 196 (60) |
| Male | 1019 (44) | 46 (26) | 639 (48) | 426 (36) | 834 (42) | 221 (46) | 936 (43) | 129 (39) |
| Other† | – | – | – | – | – | – | – | – |
| Living status, n (%) | | | | | | | | |
| Alone | 637 (27) | 58 (33) | 363 (27) | 332 (28) | 549 (28) | 140 (29) | 605 (28) | 90 (28) |
| With others | 1692 (73) | 116 (67) | 960 (73) | 848 (72) | 1437 (72) | 339 (71) | 1571 (72) | 237 (72) |
| Education programme, n (%) | | | | | | | | |
| Medical/health | 1104 (47) | 80 (46) | 708 (54) | 476 (40) | 976 (49) | 189 (40) | 1019 (47) | 165 (51) |
| Technical | 926 (40) | 63 (36) | 451 (34) | 538 (46) | 774 (39) | 202 (42) | 891 (41) | 98 (30) |
| Social sciences/ humanities | 205 (9) | 23 (13) | 105 (8) | 123 (10) | 148 (8) | 74 (15) | 193 (9) | 35 (11) |
| Economic† | – | – | – | – | – | – | – | – |
| Other† | – | – | – | – | – | – | – | – |
| Highest parental education, n (%) | | | | | | | | |
| University | 1706 (73) | 120 (69) | 967 (73) | 859 (73) | 1479 (74) | 324 (68) | 1592 (73) | 234 (72) |
| Below university | 623 (27) | 54 (31) | 356 (27) | 321 (27) | 507 (26) | 155 (32) | 584 (27) | 93 (28) |
| Study year, n (%) | | | | | | | | |
| First year | 984 (42) | 48 (28) | 507 (38) | 525 (44) | 840 (42) | 172 (36) | 919 (42) | 113 (35) |
| Above first year | 1345 (58) | 126 (72) | 816 (62) | 655 (66) | 1146 (58) | 307 (64) | 1257 (58) | 214 (65) |
| Place of birth, n (%) | | | | | | | | |
| Sweden | 1873 (80) | 122 (70) | 1050 (79) | 945 (80) | 1599 (81) | 371 (78) | 1721 (79) | 274 (84) |
| Scandinavia† | – | – | 52 (4) | 38 (3) | 77 (4) | 12 (3) | – | – |
| Europe | – | – | 83 (6) | 68 (6) | 108 (5) | 37 (8) | – | – |
| Outside Europe | 229 (10) | 38 (22) | 138 (10) | 129 (11) | 202 (10) | 59 (12) | 231 (11) | 36 (11) |

*38 persons excluded due to studying at programmes not organised in study groups.
†Some numbers are not presented due to low cell counts.

**Table 3** Characteristics of the participants in the musculoskeletal health cohort by exposure status

| | Discrimination | | High study pace | | Low social cohesion* | | Poor physical environment | |
|---|---|---|---|---|---|---|---|---|
| | Unexposed (n=2632) | Exposed (n=239) | Unexposed (n=1284) | Exposed (n=1587) | Unexposed (n=2133) | Exposed (n=690) | Unexposed (n=2434) | Exposed (n=437) |
| Age, mean (SD) | 24.5 (6.1) | 24.7 (6.1) | 24.8 (6.2) | 24.3 (6.0) | 24.4 (6.0) | 24.6 (6.0) | 24.3 (6.0) | 25.5 (6.8) |
| Gender, n (%) | | | | | | | | |
| Female | 1536 (58) | 162 (67) | 667 (52) | 1031 (65) | 1283 (60) | 384 (55) | 1426 (59) | 272 (62) |
| Male | 1084 (41) | 73 (31) | 610 (48) | 547 (35) | 840 (40) | 300 (44) | 998 (41) | 159 (36) |
| Other† | – | – | – | – | – | – | – | – |
| Living status, n (%) | | | | | | | | |
| Alone | 732 (28) | 74 (31) | 346 (27) | 460 (29) | 598 (28) | 202 (29) | 687 (28) | 119 (27) |
| With others | 1900 (72) | 165 (69) | 938 (73) | 1127 (71) | 1535 (72) | 488 (71) | 1747 (72) | 318 (73) |
| Education programme, n (%) | | | | | | | | |
| Medical/health | 1176 (45) | 95 (40) | 668 (52) | 603 (38) | 1001 (47) | 252 (37) | 1074 (44) | 197 (46) |
| Technical | 1130 (43) | 101 (42) | 461 (36) | 770 (49) | 869 (41) | 336 (49) | 1080 (44) | 151 (35) |
| Social sciences/ humanities | 214 (8) | 29 (12) | 98 (7) | 145 (9) | 156 (7) | 83 (12) | 196 (8) | 47 (11) |
| Economic† | – | – | 30 (2) | 55 (4) | – | – | – | – |
| Other† | – | – | 27 (2) | 13 (1) | – | – | – | – |
| Highest parental education, n (%) | | | | | | | | |
| University | 1935 (74) | 162 (68) | 948 (74) | 1149 (72) | 1589 (75) | 477 (69) | 1781 (73) | 316 (72) |
| Below university | 697 (26) | 77 (32) | 336 (26) | 438 (28) | 544 (26) | 213 (31) | 653 (27) | 121 (28) |
| Study year, n (%) | | | | | | | | |
| First year | 1122 (43) | 85 (36) | 491 (38) | 716 (45) | 935 (44) | 248 (36) | 1044 (43) | 163 (37) |
| Above first year | 1510 (57) | 154 (64) | 793 (62) | 871 (55) | 1198 (56) | 442 (64) | 1390 (57) | 274 (63) |
| Place of birth, n (%) | | | | | | | | |
| Sweden | 2083 (79) | 162 (68) | 1015 (79) | 1232 (78) | 1695 (80) | 520 (76) | 1891 (78) | 356 (82) |
| Scandinavia† | – | – | 47 (4) | 48 (3) | 76 (4) | 18 (3) | – | – |
| Europe† | – | – | 80 (6) | 92 (6) | 117 (6) | 50 (7) | – | – |
| Outside Europe | 303 (12) | 54 (23) | 142 (11) | 215 (14) | 245 (12) | 102 (15) | 302 (12) | 55 (12) |

*48 persons excluded due to studying at programmes not organised in study groups.
†Some numbers are not presented due to low cell counts.

**Table 4** HRs of mental health problems and activity-limiting musculoskeletal problems in any body location comparing exposed to unexposed for different aspects of a poor study environment

| | Mental health problems | | Activity-limiting musculoskeletal problems | |
|---|---|---|---|---|
| | Crude HR (95% CI) | Adjusted HR* (95% CI) | Crude HR (95% CI) | Adjusted HR* (95% CI) |
| Discrimination | 1.89 (1.52 to 2.35) | 1.75 (1.40 to 2.19) | 1.43 (1.16 to 1.77) | 1.39 (1.12 to 1.72) |
| High study pace | 1.83 (1.61 to 2.09) | 1.70 (1.48 to 1.94) | 1.21 (1.06 to 1.38) | 1.25 (1.09 to 1.43) |
| Low social cohesion | 1.55 (1.33 to 1.80) | 1.51 (1.29 to 1.77) | 1.06 (0.92 to 1.23) | 1.08 (0.93 to 1.25) |
| Poor physical environment | 1.17 (0.97 to 1.41) | 1.20 (0.99 to 1.45) | 1.23 (1.04 to 1.46) | 1.20 (1.01 to 1.43) |

*Adjusted for gender, age, living situation, education type, year of study, place of birth and highest level of parental education.

results (online supplemental eTable 5). Treating each response category as a separate exposure level indicated dose–response relationships between high study pace and both mental health problems and activity-limiting musculoskeletal problems as well as between low social cohesion and mental health problems (online supplemental eTable 6).

## DISCUSSION

Several aspects of the study environment were associated with the incidence of mental health problems and activity-limiting musculoskeletal problems in this sample of Swedish university students, even after adjustment for multiple potential confounders. Discrimination at university was experienced by 7%–8% of our sample and associated with the incidence of both mental health problems and activity-limiting musculoskeletal problems. Our results are in line with previous findings on discrimination and mental health[16 17] but highlights that discrimination in the university setting may also be of importance in relation to activity-limiting musculoskeletal problems. The association was somewhat weaker for activity-limiting musculoskeletal problems than for mental health problems, which is in line with earlier research suggesting that discrimination increases the risk of chronic pain by mediation from psychological distress.[18]

High study pace was commonly experienced in our sample and associated with the incidence of both mental health problems and activity-limiting musculoskeletal problems, which corroborates earlier research.[19 20] Sensitivity analyses indicated a dose–response relationship between high study pace and both mental health problems and activity-limiting musculoskeletal problems. High study pace was, like discrimination, more strongly associated with the incidence of mental health problems than with activity-limiting musculoskeletal problems.

Low social cohesion at university was experienced by about a fifth of our sample and associated with higher incidence of mental health problems, with sensitivity analyses indicating a dose–response relationship. This

association has not, to our knowledge, been studied among university students previously, but the results are in line with research showing associations between lack of school cohesion and mental health problems among adolescents.[23] Low social cohesion was also associated with higher incidence of activity-limiting musculoskeletal problems, although the association was weak, and the CIs did not exclude an effect in the opposite direction. Although inconclusive, these results are in line with earlier research from working populations.[24]

The physical environment at university was perceived as good by the majority of our sample. However, perceiving the environment as poor was associated with a somewhat higher incidence of both mental health problems and activity-limiting musculoskeletal problems, although for mental health problems the CIs s did not exclude an association in the opposite direction. These findings support earlier findings of association between physical aspects of the study environment and pain.[25 26]

Gender-stratified analyses showed that exposure to high study pace and low social cohesion may be more strongly associated to mental health problems among men than among women. However, these gender differences are uncertain due to overlapping CIs and larger studies are needed to draw firm conclusions on these potential gender differences.

Mental health problems and musculoskeletal problems are some of the most pressing public health concerns for young adults[1] and are highly prevalent among university students.[2–4] Our results indicate that aspects of the study environment may be an important target for universities trying to tackle the high levels of mental health problems and activity-limiting musculoskeletal problems among university students. Addressing poor study environment issues might to some extent reduce incidence of mental health problems and activity-limiting musculoskeletal problems among university students. Future research on the effect of intervening on these study environment factors on mental health and musculoskeletal problems is warranted.

## Strengths and limitations

This study has several strengths but also limitations. We were able to recruit a large sample of university students, which allowed us to study the associations between different study environment exposures and the outcomes with good precision. Further, the longitudinal design and the control for multiple potential confounders strengthens the level of evidence for a causal association.

There are, however, limitations hindering us from ascertaining causality. First, there is still the possibly unmeasured and residual confounding. Factors such as genetic vulnerability and neuroticism may have affected both the exposures and the outcomes and thereby confounded our estimates. We assumed that subclinical symptoms at baseline were on the causal pathway from the exposures to the outcomes. If this assumption does not hold, we have the risk of reverse causality. Our sensitivity analyses showed that all associations weakened when controlling for baseline levels of depression, anxiety and stress symptoms, as expected both if subclinical symptoms were mediators or confounders. However, all exposures, except poor physical environment, were still associated with the incidence of mental health problems, suggesting that reverse causality is unlikely to explain away the findings. For the incidence of activity-limiting musculoskeletal problems, the CIs of all exposures, except for discrimination, included the null when controlling for baseline mental health symptoms, leaving these associations more vulnerable to confounding by prior mental health problems. Second, since there was lost to follow-up in this study, our estimates may be affected by selection bias. Sensitivity analyses showed that all exposures were rather weakly associated to lost to follow-up, making it unlikely that selection bias would have affected the internal validity to any large extent.[36] Selection bias could have affected the external validity (ie, generalisability) of our results, and potentially explain away some of the weaker associations (online supplemental eTable 6), although it is hard to determine the direction of this potential bias. Third, the outcomes were measured using valid questionnaires, which limits the risk for misclassification in the outcomes. The exposures were measured using single item questions, and it is possible that misclassification of the exposures is present. We believe that this error is most likely to be non-differential, which would attenuate the true associations. However, since self-rated measures were used, the risk of differential misclassification cannot be ruled out, (eg, due to common source bias), which could lead to overestimation of the associations. The use of single item exposures also limits our ability to differentiate between different aspects of the exposures. For instance, our measure of the perceived physical environment measure is quite crude and does not differentiate between different aspects of the physical environment (eg, ergonomic workspaces, lighting, noise).

Most of the data used in this study were collected after the COVID-19 pandemic broke out. During this period, university education in Sweden was mostly delivered online. The switch to online education clearly affected the study environment, but effects on student health are uncertain.[38–40] Sensitivity analyses showed that our estimates were unaffected by adjustments for whether the exposures were measured before or during the pandemic. Still, it is possible that the mechanism linking study environment factors to mental and musculoskeletal health could have been different during the pandemic. For instance, physical study environment may have had larger effects on the outcomes, had the students been followed during a time period when they spent more time on campus. Thus, it is not certain to what extent these results, especially those regarding the physical study environment, will generalise to non-pandemic periods. Further, our sample was restricted to certain universities, and to students agreeing to participate, and does not fully represent the overall Swedish student population. It is therefore unclear how well these results may generalise to other populations. We believe, however, that the associations observed in this study are similar to those in other populations at other time periods, a belief supported by the alignment of our results to those of prior research conducted in different populations before the COVID-19 pandemic.[16–20 23–26]

## Conclusions

Several aspects of the study environment were associated with the incidence of mental health problems and activity-limiting musculoskeletal problems in our sample of Swedish university students, after controlling for multiple potential confounders.

**Author affiliations**
[1]Department of Health Promotion Science, Sophiahemmet University, Stockholm, Sweden
[2]Department of Psychology, Uppsala University, Uppsala, Sweden
[3]Unit of Intervention and Implementation Research for Worker Health, Institute of Environmental Medicine, Karolinska Institutet, Stockholm, Sweden
[4]Department of Clinical Neuroscience, Karolinska Institutet, Stockholm, Sweden
[5]Naprapathögskolan – Scandinavian College of Naprapathic Manual Medicine, Stockholm, Sweden

**Acknowledgements** We thank Pierre Côté, Klara Edlund and Tobias Sundberg for their important contributions in planning and performing the SUN-study data collection. We would also like to express our appreciation to all the participating students for their contribution to the SUN-study.

**Contributors** Conceptualisation: FJ, JB, HA and ES; Data curation: FJ; Formal analysis: FJ; Funding acquisition: ES; Investigation: FJ, CO and ES; Methodology: FJ and ES; Project administration: ES; Supervision: AHB and ES; Writing–original draft: FJ, JB, HA and ES; Writing–review and editing: FJ, JB, HA, IJ, CO, AHB and ES. FJ accepts full responsibility for the finished work and the conduct of the study, had access to the data, and controlled the decision to publish

**Funding** This research project was funded by the Swedish Research Council for Health, Working Life and Welfare (FORTE), grant number FORTE2018-00402. The SUN-study also received financial support from the Public Health Agency of Sweden.

**Competing interests** None declared.

**Patient and public involvement** Patients and/or the public were involved in the design, or conduct, or reporting, or dissemination plans of this research. Refer to the Methods section for further details.

**Patient consent for publication** Not applicable.

**Ethics approval** This study involves human participants and the study was approved by the Swedish Ethical Review Authority (2019-03276), and all participants provided informed consent electronically before entering the study. Participants gave informed consent to participate in the study before taking part.

**Provenance and peer review** Not commissioned; externally peer reviewed.

**Data availability statement** No data are available. The dataset analysed in the current study is not publicly available due to secondary confidentiality and privacy of the participants.

**ORCID iDs**
Fred Johansson http://orcid.org/0000-0001-9717-0935
Anne H Berman http://orcid.org/0000-0002-7709-0230

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
