## [Reviewer comments · BMJ Open]

ARTICLE DETAILS

TITLE (PROVISIONAL)	Study Environment and the Incidence of Mental Health Problems and Activity-Limiting Musculoskeletal Problems among University Students: The SUN Cohort Study
AUTHORS	Johansson, Fred; Billquist, Jessica; Andreasson, Hanna; Jensen, Irene; Onell, Clara; Berman, Anne H.; Skillgate, Eva

VERSION 1 – REVIEW

REVIEWER	Halonen, Jaana National Institute for Health and Welfare, Health Security
REVIEW RETURNED	07-May-2023

GENERAL COMMENTS	The paper is well-written and it uses relevant methods. The findings suggest that several study environment factors associate with students' health. However, I find that more sensitivity analyses should be conducted to confirm the findings. My major concern is that Covid-19 pandemic was not considered in the analyses or Discussion. Some of the data collection was done during the pandemic (August 2019 and December 2021), which may have affected the study location (at the university of at home?) and mental and physical health of the students. Some additional analyses to control for this and clarification of the pandemic restrictions in Sweden need to be conducted. More detailed comments and suggestions that are listed below. Abstract 1. I think the word "respectively" is unnecessary in the first sentence.2. In: "...and 2889 (68%) disabling pain at baseline." Do the authors mean that this many participants were with or without disabling pain at baseline? Please clarify.3. I assume that the exposure was measured at baseline as well. Please clarify the timing of exposure measurement. Strengths and limitations 1. Could you specify which way the associations might be biased due to misclassification and due to selection bias? Method 1) Design and participants. The flow chart is good. In addition, please give the number of eligible participants and response rate in the text (for both cohorts).2) Page 8, there seems to be an extra "in" in: "Disabling musculoskeletal pain in was..."3) Sensitivity analyses. If I understand correctly, part of the data collection (2019-2021) was done during the Covid-19 pandemic that has been linked to mental health problems among students.
---

	I'm not sure what was the policy in Sweden, did the students have on-site teaching or did they have to study remotely during the pandemic? I don't see how the pandemic was controlled for in the analyses though it might affect the findings for data collected in 2020 and 2021 (e.g., the findings regarding the physical study environment might be affected if student were at home). A variable for data collection year could be added as a covariate and possibly the analyses could be stratified by data collection year (pandemic/non-pandemic time) to assess how the pandemic affected the findings. 4) Sensitivity analyses. Job stress has been linked to depression and this association has been shown to be mediated by low back pain https://pubmed.ncbi.nlm.nih.gov/33418370/ Thus, I would suggest doing a sensitivity analyses including pain variable in the model assessing the association between study phase and mental health. 5) Sensitivity analyses. High study pace included 2/3 of the response options, 47% reporting high exposure. It would be interesting to see results to the highest vs. lowest study pace group. Results 1) There are rather clear gender differences in the exposures (Table 2, Table 3) that suggest some gender differences might occur. Did you test gender interactions in the analyses? Discussion 1) Page 17. Please discuss the possible direction of the bias due to loss-to follow-up in this study. Did you do any checks if the drop-outs differed (at baseline) from those responding e.g. 2 times? 2) As the exposures and outcomes were self-reported common source bias is also possible. Please discuss this. 3) Page 17. The sentence needs re-wording: "We believe, however, that found associations found in this study are..." Perhaps using "observe" instead of "find". 4) The role of Covid-19 pandemic possibly affecting these findings should be discussed.
--	--

VERSION 1 – AUTHOR RESPONSE

Fred Johansson

Comments to the Author:

The paper is well-written and it uses relevant methods. The findings suggest that several study environment factors associate with students' health. However, I find that more sensitivity analyses should be conducted to confirm the findings. My major concern is that Covid-19 pandemic was not considered in the analyses or Discussion. Some of the data collection was done during the pandemic (August 2019 and December 2021), which may have affected the study location (at the university of at home?) and mental and physical health of the students. Some additional analyses to control for this and clarification of the pandemic restrictions in Sweden need to be conducted.

More detailed comments and suggestions that are listed below.

Abstract

1. I think the word “respectively” is unnecessary in the first sentence.

Agreed, this has been removed.

2. In: “...and 2889 (68%) disabling pain at baseline.” Do the authors mean that this many participants were with or without disabling pain at baseline? Please clarify.

This sentence missed the word “without”, which has now been corrected.

3. I assume that the exposure was measured at baseline as well. Please clarify the timing of exposure measurement.

This has now been clarified in the abstract.

Strengths and limitations

1. Could you specify which way the associations might be biased due to misclassification and due to selection bias?

We have expanded the discussion on the potential direction of bias under the Strengths and limitations section in the Discussion. Since these biases may go in either direction depending on the structure of the bias (differential or non-differential misclassification for example), we believe that this discussion is too lengthy to place in the bullet points.

Method

1) Design and participants. The flow chart is good. In addition, please give the number of eligible participants and response rate in the text (for both cohorts).”

We have considered this but believe that including this in the text as well as the flowchart would not benefit the readability of the text.

2) Page 8, there seems to be an extra “in” in: “Disabling musculoskeletal pain in was...”

Thank you for noticing this, it has now been corrected.

3) Sensitivity analyses. If I understand correctly, part of the data collection (2019-2021) was done during the Covid-19 pandemic that has been linked to mental health problems among students. I’m not sure what was the policy in Sweden, did the students have on-site teaching or did they have to study remotely during the pandemic? I don’t see how the pandemic was controlled for in the analyses though it might affect the findings for data collected in 2020 and 2021 (e.g., the findings regarding the physical study environment might be affected if student were at home).

A variable for data collection year could be added as a covariate and possibly the analyses could be stratified by data collection year (pandemic/non-pandemic time) to assess how the pandemic affected the findings.

We fully agree with this comment and have now added a sensitivity analysis where we have added a covariate indicating if the baseline survey was collected before or after the onset of the COVID-19 pandemic. These results are presented under “Sensitivity analyses” at the end of the Results section and in the Supplemental eTable 1.

4) Sensitivity analyses. Job stress has been linked to depression and this association has been shown to be mediated by low back pain <https://pubmed.ncbi.nlm.nih.gov/33418370/> Thus, I would suggest doing a sensitivity analyses including pain variable in the model assessing the association between study phase and mental health.

We also believe that pain is a likely mediator the effects of study pace on mental health, and potentially also for the other exposures on mental health. And vice versa, that mental health may mediate the effects of the exposures on pain. However, since these mediators are on the causal pathway from the exposures to the outcomes, they will not bias our estimates. Therefore, we have decided not to include potential mediators in the models.

5) Sensitivity analyses. High study pace included 2/3 of the response options, 47% reporting high exposure. It would be interesting to see results to the highest vs. lowest study pace group.

We agree that this is interesting. We have now added dose-response analyses, where each response category is treated as a separate exposure level for both high study pace and low social cohesion in Supplemental eTable 5.

Results

1) There are rather clear gender differences in the exposures (Table 2, Table 3) that suggest some gender differences might occur. Did you test gender interactions in the analyses?

This is an important consideration and we have now included gender-stratified analyses in the Supplemental eTable 2.

Discussion

1) Page 17. Please discuss the possible direction of the bias due to loss-to follow-up in this study. Did you do any checks if the drop-outs differed (at baseline) from those responding e.g. 2 times?

We have now performed sensitivity analyses assessing the association between the exposures and loss to follow-up at the 12-month follow-up (Supplemental eTable 3), and expanded the Discussion under Strengths and limitations on how this might affect our results.

2) As the exposures and outcomes were self-reported common source bias is also possible. Please discuss this.

We fully agree with this comment, and mention this in the Discussion (under Strengths and limitations) in terms of differential misclassification. We have now also clarified that this could for instance be due to common source bias, which could lead to an overestimation of our estimates.

3) Page 17. The sentence needs re-wording: "We believe, however, that found associations found in this study are..." Perhaps using "observe" instead of "find".

Thank you for noticing this, we have now corrected the sentence.

4) The role of Covid-19 pandemic possibly affecting these findings should be discussed.

We agree and has added this to the discussion under Strengths and limitations, acknowledging the uncertainty regarding whether our results will be generalizable to more "normal" time-periods.

Reviewer: 1

Competing interests of Reviewer: No competing interests.

VERSION 2 – REVIEW

REVIEWER	Halonen, Jaana National Institute for Health and Welfare, Health Security
REVIEW RETURNED	10-Jul-2023

GENERAL COMMENTS	Thank you for the opportunity to re-review this paper on study environments and mental health and pain among university students. The paper is well-written and it has improved after the revisions. I have only a few additional comments for the authors to consider. Abstract 1. I think there is a typo in the following sentence: “The participants were followed at five time-points over on year using web surveys.” Methods 1. Paragraph “Patient and public involvement” The following is a bit unclear: “...have been offered to take part of the results of the study” Do you mean: “...take part in the dissemination of the study results”? 2. Paragraph “Exposures”. The authors write: “Participants with ratings >3 were considered exposed to discrimination)”. What was the justification for this cut-off point? I see that the prevalence of discrimination is rather low; 7-8% in the cohorts with this cut-off. Was the cut-off based on prevalence in the data? Please clarify the justification for the cut-off. 3. Paragraph “Sensitivity analyses”. I would remove references to supplemental result tables from this section and only refer to those in the Results section as the results are not presented here. Results 1. eTable 4. The table title says E-values are provided, but the column headings have “RR (95% CI)”. I am not sure if the E-values are presented as Risk Ratios so perhaps some clarification on this could be added either in the table or the Methods part, or both. 2. The following sentence is a bit unclear: “The main results were not confounded by if the exposures were measured before or during the COVID-19 pandemic” Please revise. 3. When talking about the gender differences it would be good to mention in the text (not all readers will check the supplemental tables) that the mental health effects might be stronger among men than women (for all exposures the mental health effect estimates were higher for men than women). Discussion 1. The gender differences are shortly discussed mentioning the larger effect estimates for men. Although the interaction was not significant (overlapping CIs) it would be interesting to discuss shortly possible reasons for why the effect seemed to be stronger
---

	among men, particularly while the exposure prevalence was higher among women.
--	---

VERSION 2 – AUTHOR RESPONSE

Reviewer: 1

Dr. Jaana Halonen, National Institute for Health and Welfare, Stockholm University

Comments to the Author:

Abstract

1. I think there is a typo in the following sentence: “The participants were followed at five time-points over on year using web surveys.”

Thank you for noticing this, this has now been corrected.

Methods

1. Paragraph “Patient and public involvement” The following is a bit unclear: “...have been offered to take part of the results of the study” Do you mean: “...take part in the dissemination of the study results”?

We agree that this was not entirely clear, we have now changed to “and some have participated in seminars on the results, organized by the research group”.

2. Paragraph “Exposures”. The authors write: “Participants with ratings >3 were considered exposed to discrimination)”. What was the justification for this cut-off point? I see that the prevalence of discrimination is rather low; 7-8% in the cohorts with this cut-off. Was the cut-off based on prevalence in the data? Please clarify the justification for the cut-off.

Thank you for noticing this need for clarification. We chose to limit the exposure to those indicating a severity level of >3, which included the top 50 % of the severity ratings, in order to identify more severe forms of discrimination. This was decided a-priori, since we believed that the more severe forms of discrimination would be of most importance to mental and musculoskeletal health. We have now added this justification under the paragraph “Exposures”.

3. Paragraph “Sensitivity analyses”. I would remove references to supplemental result tables from this section and only refer to those in the Results section as the results are not presented here.

Agreed, references to the Supplemental Tables are now removed from the Methods section.

Results

1. eTable 4. The table title says E-values are provided, but the column headings have “RR (95% CI)”. I am not sure if the E-values are presented as Risk Ratios so perhaps some clarification on this could be added either in the table or the Methods part, or both.

Thank you for noticing this need for clarification. The E-values represents the minimum

RR between missingness and the outcome needed to shift the point estimates to the

null. So, the E-values are measured on the risk ratio scale. This has now been clarified in the Methods section.

2. The following sentence is a bit unclear: “The main results were not confounded by if the exposures were measured before or during the COVID-19 pandemic” Please revise. Agreed, this has now been changed to: “Adding a variable indicating if exposures were measured before or during the COVID-19 pandemic did not affect the main results”.

3. When talking about the gender differences it would be good to mention in the text (not all readers will check the supplemental tables) that the mental health effects might be stronger among men than women (for all exposures the mental health effect estimates were higher for men than women).

We have now expanded a bit on potential gender differences in the results section. It now reads:

“Gender stratified analyses showed some potential differences in the strength of the associations between men and women for mental health, although differences are uncertain due to overlapping confidence intervals. The point estimates for low social cohesion and high study pace were higher among men (Supplemental eTable 3).”

Discussion

1. The gender differences are shortly discussed mentioning the larger effect estimates for men. Although the interaction was not significant (overlapping CIs) it would be interesting to discuss shortly possible reasons for why the effect seemed to be stronger among men, particularly while the exposure prevalence was higher among women.

We believe that these results are still too uncertain to warrant a discussion on potential reasons. We have added the sentence "larger studies are needed to draw firm conclusions on these potential gender differences" to highlight this.